# MALDI-TOF MS Identification of Dromedary Camel Ticks and Detection of Associated Microorganisms, Southern Algeria

**DOI:** 10.3390/microorganisms10112178

**Published:** 2022-11-03

**Authors:** Fatima Zohra Hamlili, Maureen Laroche, Adama Zan Diarra, Ismail Lafri, Brahim Gassen, Boubaker Boutefna, Bernard Davoust, Idir Bitam, Philippe Parola

**Affiliations:** 1IHU Méditerranée Infection, 19-21 Boulevard Jean Moulin, 13005 Marseille, France; 2Aix Marseille Univ, IRD, AP-HM, SSA, VITROME, 13005 Marseille, France; 3Laboratoire des Biotechnologies Liées à la Reproduction Animale, Institut des Sciences Vétérinaires, Université de Blida 1, Blida 09000, Algeria; 4Laboratoire Mixte International REMEDIER, VITROME, IRD, 13005 Marseille, France; 5Etablissement Public Hospitalier de Tamanrasset, Tamanrasset 11008, Algeria; 6Inspection Vétérinaires, Direction des Services Agricoles, Tamanrasset 11008, Algeria; 7Aix Marseille Univ, IRD, AP-HM, MEPHI, 13005 Marseille, France; 8Ecole Supérieure des Sciences de l’Aliment et des Industries Alimentaires, Alger 16000, Algeria

**Keywords:** MALDI-TOF MS, ticks, *Hyalomma*, camels, *Anaplasma platys*, *Coxiella burnetii*, Algeria

## Abstract

This study used MALDI-TOF MS and molecular tools to identify tick species infesting camels from Tamanrasset in southern Algeria and to investigate their associated microorganisms. Ninety-one adult ticks were collected from nine camels and were morphologically identified as *Hyalomma* spp., *Hyalomma dromedarii*, *Hyalomma excavatum*, *Hyalomma impeltatum* and *Hyalomma anatolicum*. Next, the legs of all ticks were subjected to MALDI-TOF MS, and 88/91 specimens provided good-quality MS spectra. Our homemade MALDI-TOF MS arthropod spectra database was then updated with the new MS spectra of 14 specimens of molecularly confirmed species in this study. The spectra of the remaining tick specimens not included in the MS database were queried against the upgraded database. All 74 specimens were correctly identified by MALDI-TOF MS, with logarithmic score values ranging from 1.701 to 2.507, with median and mean values of 2.199 and 2.172 ± 0.169, respectively. One *H. impeltatum* and one *H. dromedarii* (2/91; 2.20%) tested positive by qPCR for *Coxiella burnetii*, the agent of Q fever. We also report the first detection of an *Anaplasma* sp. close to *A. platys* in *H. dromedarii* in Algeria and a potentially new *Ehrlichia* sp. in *H. impeltatum*.

## 1. Introduction

Ticks are obligate hematophagous ectoparasites that infest most vertebrates worldwide, from the warmest regions of the globe to the coldest ones [1]. To date, over 900 tick species have been described around the world [2]. Ticks are significant vectors of a wide range of microorganisms, including bacteria, helminths, viruses and protozoa [1,3,4,5]. They are considered a major global human and veterinary public health problem by being reservoirs and/or vectors of infectious disease agents to susceptible hosts [6].

Dromedary camel (*Camelus dromedarius*) populations are widespread in the Middle East and North Africa, including Algeria [7,8,9,10]. They play an important role in the culture and economy of some countries. In fact, they represent their main source of milk and meat production because of their very unique adaptative physiological characteristics to arid and semi-arid ecosystems [11,12]. Camel production is negatively impacted by zoonotic diseases [7,13]. Ticks are one of the most predominant ectoparasites that negatively affect their productivity as well as their performance by either transmitting infectious pathogenic agents or causing traumatic lesions and severe anemia due to blood loss [9,14,15].

Camels can be infested with different tick species, such as *Hyalomma dromedarii*, *H. impeltatum*, *H. excavatum*, *H. marginatum*, *H. anatolicum*, *H. impressum*, *H. truncatum*, *Rhipicephalus sanguineus*, *R. pulchellus*, *Amblyomma variegatum* and *A gemma*. Camel ticks have been associated with several tick-borne pathogens, including *Babesia* spp., *Rickettsia* spp., *Anaplasma* spp., *Coxiella burnetii*, *Ehrlichia* spp. and *Theileria* spp. [7,16,17,18,19]. *Anaplasma phagocytophilum* and *A. marginale* have been implicated in anaplasmosis in camels in India, Iran, Saudi Arabia and Tunisia [20,21,22,23,24]. In Algeria, a molecular investigation of Anaplasma species in dromedaries and their ticks is lacking. Nevertheless, *C. burnetii*, the agent of Q fever, and *Rickettsia africae*, the agent of African tick bite fever, were detected in camel *Hyalomma* spp. ticks from Algeria [17,25].

Tick identification is essential for the epidemiological mapping of vectors and tick-borne diseases. Accurate identification is important for effective control strategies [1,6]. Tick characterization is mainly based either on their morphology or on the use of DNA-based methods [26]. Nevertheless, the morphological identification of ticks still requires tremendous entomological expertise. In some cases, the identification criteria can be ambiguous in immature stages (larvae and nymphs), damaged or engorged specimens, and cryptic species. The method is also limited by the availability of relevant taxonomic keys. As for molecular methods, they are laborious, expensive and usually conditioned by the choice of the relevant target gene and by the availability of good-quality sequences in GenBank [27,28].

Recently, matrix-assisted laser desorption/ionization time-of-flight mass spectrometry (MALDI-TOF MS) has been used in medical entomology for the identification of many arthropods of medical and veterinary importance [27,29,30,31]. This proteomic tool has been successfully used in identifying tick species by analyzing the proteins extracted from their legs [27,32,33]. 

Our study aimed to test the ability of MALDI-TOF MS to identify tick species infesting dromedary camels in Algeria and to detect associated microorganisms using molecular tools. 

## 2. Materials and Methods

### 2.1. Ethics Statement

Risk assessment was submitted to and approved by the ethics committee and decision board of the Inspection Vétérinaire (Direction des Services Agricoles) of Tamanrasset province of Algeria. This institution is affiliated with the Directions des Services Veterinaires (Algerian Ministry of Agriculture and Rural Development). To facilitate fieldwork, collaboration was established with the veterinary inspector of Tamanrasset province. The dromedary camels were handled by local veterinarians. Importation authorization was obtained from the relevant entity (DDPP/Prefecture of Bouches-du-Rhône, Marseille, France) in August 2018 under the number ER 11/18.

### 2.2. Study Area and Tick Sampling

In August 2018, nine camels were examined for tick collection at the livestock market in the commune of Tamanrasset, which is located in the province of Tamanrasset, in southernmost Algeria, at the border with Mali and Niger (22°47′25.069′′ N, 5°31′9.577′′ E) (Figure 1). Ticks were carefully removed using forceps and placed in individually labeled tubes for each camel. The tubes contained 70% ethanol and were stored at room temperature. Three months later, samples were transported to Marseille, France, for subsequent analysis. Tick morphological identification was performed by using standard keys and descriptions with a Leica EZ4 binocular microscope [34,35]. Specimens were codified according to the genus, species and sex. 

ArcGIS 10.3 software (http://www.esri.com, accessed on 25 October 2021) was used to map the spatial localization of tick sampling in our study region. 

All specimens were then subjected to MALDI-TOF MS analyses after being morphologically identified. For some specimens, molecular identification was also carried out as described below. 

### 2.3. Sample Preparation for MALDI-TOF MS

Ethanol-preserved ticks were rinsed with distilled water and then dried with filter paper. Four legs from only one side were dissected using a sterile surgical blade used for MALDI-TOF MS analysis. The leg sample preparation was performed using a previously described protocol [27]. One microliter of the supernatant of the protein extract of each sample was spotted in quadruplicate onto a MALDI-TOF plate (Bruker Daltonics, Wissembourg, France). All spots were left to dry and then covered with 1 μL of HCCA matrix composed of saturated α-cyano-4-hydroxycynnamic acid (Sigma, Lyon, France), 50% acetonitrile, 2.5% trifluoroacetic acid (Aldrich, Dorset, UK) and HPLC grade water [27]. The target plate was left to dry at room temperature and then introduced into a Microflex LT MALDI-TOF Mass Spectrometer device (Bruker Daltonics, Bremen, Germany) for analysis. 

The tick’s remaining body was longitudinally cut into two halves, one half being used for molecular identification and the detection of microorganisms and the other half being preserved at −20 °C [27].

### 2.4. Molecular Identification of Ticks

Molecular identification was performed in order to validate the morphological identification of specimens whose MS spectra were added to our in-house arthropod spectra MS database. It was also carried out to confirm the identification assigned by MALDI-TOF MS of randomly selected specimens morphologically identified as *Hyalomma* spp. 

We followed the same DNA extraction protocol described elsewhere [27]. DNA obtained from each tick specimen was subjected to standard PCR in an automated DNA thermal cycler (Applied Biosystems, 2720, Foster City, CA, USA) to amplify a 455 base-pair fragment of the *16S rRNA* gene using tick 16S primers (F-CCGGTCTGAACTCAGATCAAGT and 16S-R CTGCTCAATGATTTTTTAAATTGCTGTGG) [36]. The amplified products were purified and sequenced as described previously [32]. The obtained sequences were assembled and corrected by Chromas Pro1.77 (Technelysium Pty. Ltd., Tewantin, Australia) and a BLAST query was performed against the NCBI GenBank database (http://www.ncbi.nlm.nih.gov/blast/ accessed on 13 May 2021).

### 2.5. Microorganism Detection

All tick specimens were screened for the presence of *Rickettsia* spp., *Bartonella* spp., Anaplasmataceae spp., *Borrelia* spp., *Coxiella burnetii* and Piroplasmida using previously described primers and probes [37,38,39,40,41,42]. qPCR was performed using the CFX96 Real-Time System (Bio-Rad, Marnes-La-Coquette, France). We used positive controls from our laboratory cultures (DNA from *Rickettsia montanensis*, *Bartonella elizabethae*, *Anaplasma phagocytophilum*, *Coxiella burnetii*, *Borrelia crocidurae* and *Babesia vogeli*) and negative controls (qPCR mix without any DNA) for each qPCR plate. For *C. burnetii* qPCR, two intergenic regions were targeted *IS30A* and *IS1111*. Only samples that tested positive for both genes were considered positive. The tick specimens were considered positive when the cycle threshold (Ct) value was <36 [43]. Anaplasmataceae-positive samples were then subjected to standard PCR using 3 different primers for species determination: *16S rRNA* (Ehr-16S-D GGTACCYACAGAAGAAGTCC and Ehr-16S-R TAGCACTCATCGTTTACAGC) [44], *23S rRNA* (Ana23S-212f ATAAGCTGCGGGGAGTTGTC and Ana23S-753r TGCAAAAGGTACGCTGTCAC) [37] and *groEL* (Ehr-groEL-F GTTGAAAARACTGATGGTATGCA and Ehr-groEL-R ACACGRTCTTTACGYTCYTTAAC) [45]. The obtained sequences were assembled, corrected by Chromas Pro (Technelysium Pty. Ltd., Tewantin, Australia) and then compared with the sequences available in GenBank. The phylogenetic tree reconstruction was performed using MEGA 7 using the maximum likelihood and the model selected by MEGA 7 [46]. Bootstrap analyses were conducted using 500 replicates.

### 2.6. Spectral Analysis and Database Creation

The reproducibility of the MS profiles of tick species was assessed using both Flex analysis V 3.3 and ClinProTools v.2.2 software packages (Bruker Daltonics, Germany) [32]. Intra-species reproducibility and inter-species specificity were validated by comparing and analyzing the spectral profiles obtained from the four spots of each individual specimen. Poor-quality spectra were excluded from the analysis (<3000 arbitrary units (a.u.) and background noise). Cluster analysis (MS dendrogram) was performed with MS spectra of specimens of each species using MALDI-Biotyper 3.0 software. The database was then created by adding high-quality MS spectra of each species to our homemade arthropod spectra MS database after molecular confirmation.

### 2.7. Blind Test

All MS spectra obtained from tick leg specimens were subjected to blind test analysis, except for those used to create the MS database (MS reference spectra). The log-score values (LSVs), ranging from 0 to 3, were computed by the MALDI-Biotyper v.3.0 (Bruker Daltonic, Germany). The LSVs refer to the degree of homology between MS reference spectra and MS spectra used for the blind test [47]. 

## 3. Results

### 3.1. Morphological Identification of Ticks

Overall, 91 adult ticks were collected from 9 examined camels. These included 41 (45.05%) males and 50 (54.95%) females, all belonging to the *Hyalomma* genus. Engorged females (*n* = 41; 45.05% of all ticks) were only identified at the genus level. For those identified at the species level, the most abundant tick species were *H. dromedarii* (*n* = 42; 46.15%), followed by *H. excavatum* (*n* = 4; 4.4%) and *H. impeltatum* (*n* = 3; 3.3%). One *H. anatolicum* (1.1%) was also identified (Table 1). 

### 3.2. MS Identification of Dromedary Camel Ticks

A total of 91 tick leg samples were subjected to MALDI-TOF MS. The visualization of all MS spectra using Flex analysis v3.4 software showed that 96.70% (88/91) of MS spectra were of excellent quality. Representative MS profiles of the three tick species are presented in Figure 2A. Cluster analysis (Figure 2B) revealed inter-species specificity, as all specimens of the same species were clustered on the same branch. 

This was confirmed by a principal component analysis (PCA) diagram generated by ClinProTools software, which revealed a visible distinction between the three species (Figure 3A). 

Our homemade MALDI-TOF arthropod spectra database (MALDI-Biotyper 3.0) was upgraded with the MS reference spectra of 14 specimens, 12 *H. dromedarii*, 1 *H. excavatum* and 1 *H. impeltatum*, all of which were molecularly identified. The remaining 74 good-quality spectra from tick legs were subjected to a blind test against our upgraded database (Figure 4). The results showed that of the 36 specimens of ticks morphologically identified as *Hyalomma* spp., 34 were identified by MALDI-TOF MS as *H. dromedarii*, with LSVs ranging from 1.890 to 2.507, and two were identified as *H. excavatum*, with LSVs ranging from 1.801 to 1.880 (Table 1). For those morphologically identified as *H. dromedarii*, all were identified by MALDI-TOF MS as *H. dromedarii* as well, with LSVs ranging from 1.701 to 2.431. Those morphologically identified as *H. impeltatum* were identified by MS as *H. impeltatum*, with LSVs ranging from 2.081 to 2.083. The three ticks morphologically identified as *H. excavatum* were identified by MS as *H. excavatum*, with LSVs of 2.046–2.204 (Table 1) (Figure 4). 

The tick whose morphological identification was corrected by sequencing was identified by MS as *H. dromedarii* [LSV: 2.342], in agreement with the molecular identification (Table 1). The median of all log-score values was 2.199, and the mean LSV was 2.172 ± 0.169. All of the obtained score values of each species, *H. dromedarii*, *H. impeltatum* and *H. excavatum*, are presented in Figure 3B. Our camel tick MALDI-TOF MS database is publicly accessible and can be downloaded with the following DOI number: https://doi.org/10.35081/srfb-n029, accessed on 25 October 2021.

### 3.3. Molecular Identification of Ticks

In total, 17 specimens morphologically identified as *H. dromedarii* (*n* = 8), *Hyalomma* spp. (*n* = 4), *H. excavatum* (*n* = 3), *H. impeltatum* (*n* = 1) and *H. anatolicum* (*n* = 1) were submitted to sequencing for the MS database creation (Figure 4). BLAST analyses of ticks morphologically identified as *H. dromedarii*, *H. impeltatum* and *H. excavatum* were 99.52–100% (MN960589), 99.75% (MN960583) and 100% (KU130429/MK601704), respectively, identical to their homologous sequences available in GenBank. The sequence of the tick morphologically identified as *H. anatolicum* was in fact 100% identical to the sequence of *H. dromedarii* (MN960589).

The sequences of four randomly chosen specimens morphologically identified as *Hyalomma* spp. were 99.76–100% identical to *H. dromedarii* sequences (LC654693/MN960589) (Table 1). Finally, only the MS spectra of 14 molecularly confirmed ticks (*H. dromedarii* (*n* = 12), *H. excavatum* (*n* = 1) and *H. impeltatum* (*n* = 1)) were added to our homemade MS database.

Ten tick specimens morphologically identified as *Hyalomma* spp., of which eight were identified by MALDI-TOF MS as *H. dromedarii* and two were identified as *H. excavatum*, were also submitted to molecular analysis to verify the MS identifications (Figure 4). The sequencing results revealed that eight specimens were 99.73–100% identical to *H. dromedarii* (MN960589), and two specimens were 100% identical to *H. excavatum* (MK601704). Molecular tools confirmed the identification of ticks that were unable to be morphologically identified to the species level. The 16S sequences of the ticks obtained in this study have been deposited in the GenBank database under the following accession numbers: OL672219, OL672220, OL672221, OL672222, OL672223 and OL672224 for *H. dromedarii*, OL672225 for *H. impeltatum* and OL672226 for *H. excavatum*.

### 3.4. Microorganism Screening 

A total of 10/91 (10.98%) ticks tested positive by qPCR for microorganisms, including *C. burnetii* (2/91; 2.20%) and Anaplasmataceae spp. (8/91; 8.80%). The two ticks positive for *C. burnetii* using both the *IS30A* and *IS1111* sequences were one *H. impeltatum* and one *H. dromedarii*. Anaplasmataceae spp. were detected in two *H. impeltatum* and six *H. dromedarii* (Figure 4). No *Borrelia* spp., *Piroplasma* spp., *Rickettsia* spp. or *Bartonella* spp. were detected.

The sequencing of a fragment of the Anaplasmataceae *16rRNA* gene was only successful for three out of eight specimens. BLAST analysis revealed that the two sequences obtained from two positive *H. dromedarii* ticks showed 100% similarity to *Candidatus* Anaplasma camelii (MT510533) and *A. platys* (MN630836), respectively. One sequence detected in the *H. impeltatum* tick revealed 99.63% similarity to *Candidatus* Midichloria mitochondrii (endosymbiotic bacterium), which was previously detected in an *H. dromedarii* tick from Tunisia (MK416236) (Table 2). The maximum likelihood (ML) phylogenetic tree based on the *16S rRNA* gene (Figure 5) showed that *Anaplasma* detected in *H. dromedarii* belonged to the cluster of Ca. A. camelii and *A. platys*.

BLAST of the *23S rRNA* gene sequences obtained from these three Anaplasmataceae-positive samples showed that two sequences from *H. dromedarii* ticks were 99.78–100% identical to an uncultured *Anaplasma* sp. (MN626401), and the sequence obtained from *H. impeltatum* was 97.28% identical to *Candidatus* Ehrlichia rustica (KT364330) (Table 2). The *23S* sequence of Ca. *A. camelii* was not available in GenBank. The ML phylogenetic tree based on the *23S rRNA* gene (Figure 6) revealed that the two sequences from the *Anaplasma*-positive samples (*H. dromedarii*) belonged to the main cluster of *A. platys*. More specifically, the obtained sequences formed a separate and well-supported subcluster with the uncultured *Anaplasma* (referred to as *A. platys-like*) isolated from the blood of camels from Egypt (MN626401) but closely related to the subcluster of *A. platys*. One sequence of *Ehrlichia* sp. (*n* = 1; 1.10%) obtained from *H. impeltatum* formed an independent subcluster. However, it was close to the subcluster of *E. canis*.

The *H. impeltatum* specimen positive for Anaplasmataceae, whose sequence was close to “*Candidatus* Ehrlichia rustica” for the *23S rRNA* gene, was then subjected to sequencing with the *groEL* gene for confirmation. BLAST analysis showed that the obtained *groEL* sequence was 94.59% identical to *E. canis* (Table 2). The *groEL* ML phylogenetic analysis (Figure 7) revealed that the sequence of *Ehrlichia* sp. (*H. impeltatum*) formed an independent and well-supported (bootstrap support ≈ 100%) branch separated from the other *Ehrlichia* spp., showing that it may potentially be a novel species of *Ehrlichia*. 

In summary, two ticks, one *H. impeltatum* and one *H. dromedarii*, were positive for *C. burnetii* using both *IS30A* and *IS1111* primers, confirming the presence of *C. burnetii* DNA. In two other *H. dromedarii* (2.20%), we identified an *Anaplasma* sp. closely related to *A. platys* using the *16S* and *23S* genes and a potentially new *Ehrlichia* sp. using *23S* and *groEL* in *H. impeltatum*. The sequences of the ticks and microorganisms have been deposited in GenBank, and the accession numbers of these sequences are OL672236, OL672237 and OL672238 for the *16SrRNA* gene, OL672216, OL672217 and OL672218 for the *23S rRNA* gene and OL757650 for the *groEL* gene.

## 4. Discussion

In recent years, MALDI-TOF MS has been used to accurately and rapidly diagnose bacterial species and identify fungi [48,49]. This method is now incorporated into diagnostic routines in many clinical laboratories [50]. MALDI-TOF MS has also been used in medical entomology to identify several species of arthropods, such as ticks, mosquitoes, bed bugs, fleas, sandflies and others, using protein extracts from different body parts [28,31,33,51,52]. More recently, MALDI-TOF MS has also been used in malacology [53]. Once the device has been purchased for a technological platform, there is no additional cost for its use because this technique is inexpensive, simple and fast in comparison to conventional methods [26]. As previously mentioned, the robustness of MALDI-TOF MS for arthropod identification has been evaluated for several years using laboratory-reared and field-collected samples. MALDI-TOF MS profiling uses a highly standardized protein extraction method coupled with a spectrum acquisition approach to generate reference spectra that will be specific and reproducible enough to identify specimens of the same species [54]. MALDI-TOF MS has especially shown its reliability and usefulness in field studies by significantly decreasing the costs and time of identification but also allowing the detection of low-occurrence species [55]

Camels can be infested with different tick species of various genera [9,10,14,15]. In our study, the ticks belonged to the genus *Hyalomma*. The morphological identification of engorged females was only possible at the genus level because some morphological criteria were distorted and not visible, making their identification challenging. In our study, three species, namely, *H. dromedarii*, *H. impeltatum* and *H. excavatum*, were identified using molecular, morphological and proteomic tools. Similar observations have been reported by other researchers on these tick species infesting camels in Algeria, Tunisia and other parts of the world [10,18,56,57,58]. The camel tick *H. dromedarii* was the most frequent tick species found on camels. This species is known to have a host preference for dromedary camels but can also infest sheep, goats and horses, and it adapts well to extreme dryness and desert climates [34]. However, another study in Tunisia showed that *H. impeltatum* was the most abundant in *Camelus dromedarius*, followed by *H. dromedarii*, as described in [59]. 

In order to upgrade our database with MS spectra, as well as to randomly verify MS identifications, we used the *16S rRNA* tick gene for the sequencing and definitive identification of ticks. We obtained a discrepancy between morphological and molecular/MALDI-TOF MS identifications for one tick specimen, with the MS identification in agreement with molecular identification. As previously stated, the taxonomy of the genus *Hyalomma* includes several species with important within-species morphological variability, which makes their morphological identification challenging, even for tick experts [60,61]. The morphological error may also be due to the invisible characters of this specimen. We therefore relied on molecular tools as well as MALDI-TOF MS to overcome morphological errors and ambiguous identification. 

The MALDI-TOF MS results were shown to be in agreement with all molecular identifications. As for the ten specimens morphologically designated as *Hyalomma* spp., MALDI-TOF MS enabled the correct identification of those ticks at the species level. All ticks were identified by MALDI-TOF MS with high LSVs. This study highlights the ability of MALDI-TOF MS to overcome the limitations related to ticks’ taxonomical identification, such as the identification of damaged or engorged specimens and the lack of taxonomy keys, suitable documentation and entomological expertise [26,32]. Walker et al. (2003) reported that the accurate identification of *H. dromedarii* and *H. impeltatum* is confusing, especially those in the engorged state [34]. Here, the MS dendrogram and PCA results show that MALDI-TOF MS can discriminate between these two species, regardless of the engorged state. 

Molecular tools, including polymerase chain reaction and sequencing, have enabled a major advance in the study of zoonotic pathogens and their reservoirs. In this work, we were able to detect *C. burnetii* in ticks. This intracellular Gram-negative bacterium is the causative zoonotic agent of Q fever. However, certain ruminants are considered the principal reservoirs for the transmission of *C. burnetii* to humans. *Coxiella burnetii* transmission to humans can be through aerosol inhalation or the ingestion of infected animal products, such as unpasteurized or insufficiently pasteurized milk products or derivatives [7]. Transmission can occur through birth products, including the placenta and amniotic liquid. These represent a major risk to veterinarians and livestock or dairy farmers [62]. This bacterium has been detected in more than 40 hard and at least 14 soft tick species, but they are considered to play only a secondary role in the transmission to humans [63]. In Algeria, *C. burnetii* has been detected in ticks (*Rhipicephalus* and *Hyalomma*) and in the blood of small ruminants [10,64] and camels [10], but only one human case was reported in 2012 in the northwest of the country [65]. *Coxiella burnetii* has also been found in camels and their ticks in several countries [7,66,67]. The long-term persistence of *C. burnetii* is possible in camels, and this may pose a threat to public health as well as breeding farms. However, the role of camels in the transmission of *C. burnetii* should be further evaluated [7]. In our survey, *C. burnetii* was amplified in *H. impeltatum* and *H. dromedarii*. However, a recent study on *C. burnetii* isolated it from *H. impeltatum*, *H. dromedarii* and *H. excavatum* in Algeria [10]. In Tunisia, *C. burnetii* was reported in *H. impeltatum* and *H. dromedarii*, and in Egypt, *H. dromedarii* was positive for *C. burnetii* [68,69]. 

In our study, we report the first detection in Algeria of an *Anaplasma sp*. in *H. dromedarii*, and the species is closely related to canine *A. platys*. A potential new *Ehrlichia* sp. was also detected in *H. impeltatum*. Bastos et al. [70] reported a novel species genetically close to *A. platys* in camels and named it “*Candidatus* Anaplasma camelii”. Other studies have recorded Ca. A. camelii in dromedary camels from Saudi Arabia [70], Tunisia, Morocco [20], Nigeria [18], Kenya [71] and Iran [72]. Recently, the occurrence of “*Candidatus* Anaplasma camelii”was reported in cattle and deer in Malaysia [73] and camel keds (*Hippobosca camelina*) [71]. The prevalent variations are likely to be the result of differences in the distribution of vectors in camel-sampling localities, reservoir hosts or tick control programs [56]. In Xinjiang, China, Li et al. (2015) reported the presence of canine *A. platys* in *Camelus bactrianus* (two-humped camels) and their ticks [74]. However, this strain was genetically divergent from the *Anaplasma* strain infecting dromedaries. 

In order to characterize the obtained Anaplasmataceae species, we amplified fragments of three genes, one of which (*groEL*) is specific to *Ehrlichia*. The phylogenetic tree analysis based on the *16S rRNA* gene revealed that the *Anaplasma* strain from *H. dromedarii* in our study clustered with “*Candidatus* Anaplasma camelii” from the dromedary camel of Kenya and *A. platys* from dog blood. To see whether the obtained sequences were identical to “*Candidatus* Anaplasma camelii”or *A. platys* and to differentiate the two *Anaplasma* strains, we targeted the *23S rRNA* gene because of its ability to identify a large panel of Anaplasmataceae. As reported, this marker is ideal for discriminating between two closely related *Anaplasma* species and potentially detecting new species [37]. The phylogenetic analysis revealed that the obtained sequences formed an independent subcluster belonging to *A. platys*. In addition, the partial *23S rRNA* sequence of “*Candidatus* Anaplasma camelii”does not exist in GenBank. Indeed, the *23S rRNA* gene was able to separate *Anaplasma* sequences isolated from *H. dromedarii* and *A. platys* sequences. Consequently, we followed the proposition of *Bastos* et al. on the status of “*Candidatus* Anaplasma camelii”. 

Furthermore, phylogenetic analysis based on *23S rRNA* revealed that the sequence of *Ehrlichia* sp. detected in *H. impeltatum* formed an independent branch close to the subcluster of *E. canis*. The phylogenetical analysis based on *groEL* and a gene specific for *Ehrlichia* showed that the sequence of *Ehrlichia* formed an independent branch and was separate from other species of *Ehrlichia*. Previously, Bastos et al. confirmed the presence of a novel *Ehrlichia* species that is genetically close to *E. canis*, and another study revealed *E. canis* infections in camels in Saudi Arabia [16,70]. Just recently, a study reported the presence of an *Ehrlichia* sp. genetically related to *E. canis*, *E. regneryi* and *E. ruminantium* isolated from the blood of sick camels and ticks (*Amblyomma* spp. and *Rhipicephalus* spp.) in Kenya [75]. The occurrence of canine pathogens in camels may be due to the close association and cohabitation of camels and livestock in the desert area, which amplifies the chances of spreading vector-borne diseases [16,71]. This may also be due to the fact that some dog ticks and camel ticks have relatively low host specificity and/or are quite aggressive ticks. In addition, high temperatures make ticks particularly aggressive and further decrease host specificity. Nevertheless, further investigations are required to clarify the veterinary and medical importance of the novel species.

## 5. Conclusions

Our study showed that MALDI-TOF MS is a useful tool for the rapid identification of tick species stored in ethanol and that it could be used to overcome morphological limitations. We report the first detection of an *Anaplasma* sp. close to *A. platys* in *H. dromedarii* in Algeria and a new *Ehrlichia* sp. in *H. impeltatum*. *Ehrlichia* sp. and *Anaplasma* sp., genetically related to *A. platys* detected in camels, have not been characterized yet. However, further studies are necessary to clarify their potential zoonotic and veterinary risks. Additionally, we detected *C. burnetii* in camel ticks (*H. dromedarii* and *H. impeltatum*), as previously reported in Algerian camel ticks. In Algeria, camels and ticks could be a significant reservoir or source for the transmission of Q fever to animals and humans.

## Figures and Tables

**Figure 1 microorganisms-10-02178-f001:**
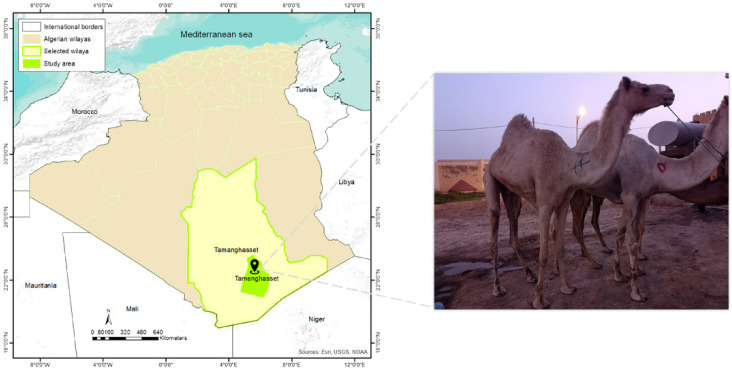
Map of Algeria showing the location of tick collection (Tamanrasset) and the examined dromedary camels. ArcGIS 10.3 software (http://www.esri.com, accessed on 25 October 2021).

**Figure 2 microorganisms-10-02178-f002:**
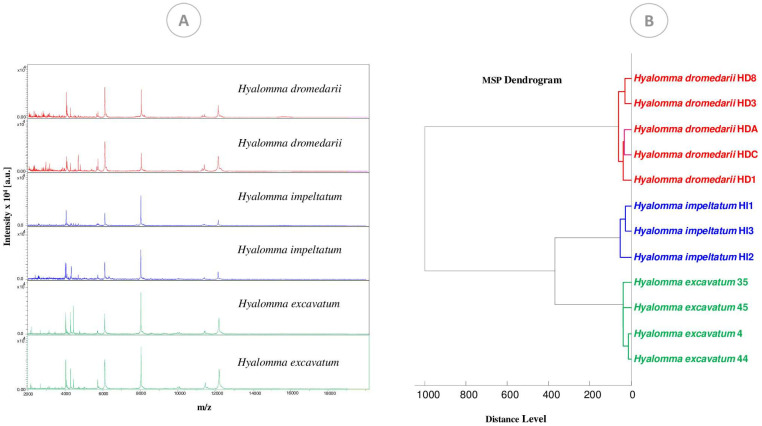
MALDI-TOF MS spectra obtained from tick species added to the database: *H. dromedarii*, *H. impeltatum* and *H. excavatum*. (**A**) Spectral alignment of the three species using Flex Analysis software and (**B**) MS dendrogram created using MS spectra from the three species using Biotyper software v.3.0.

**Figure 3 microorganisms-10-02178-f003:**
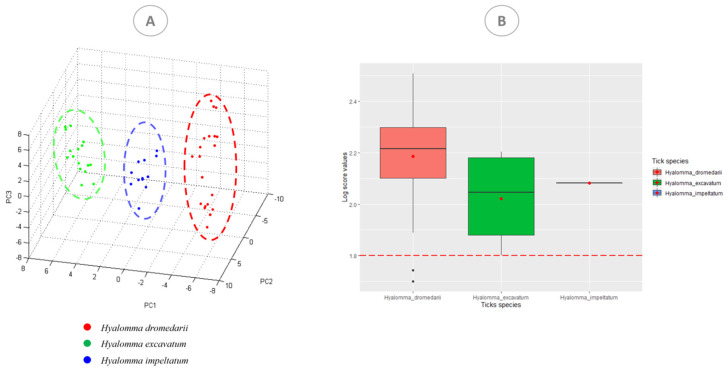
MALDI-TOF MS spectra obtained from tick species. (**A**) Comparison of MALDI-TOF MS spectra of three species by principal component analysis using ClinProTools 2.2. (**B**) Graphical representation showing the log-score value classification according to the present species: *H. dromedarii*, *H. impeltatum* and *H. excavatum*.

**Figure 4 microorganisms-10-02178-f004:**
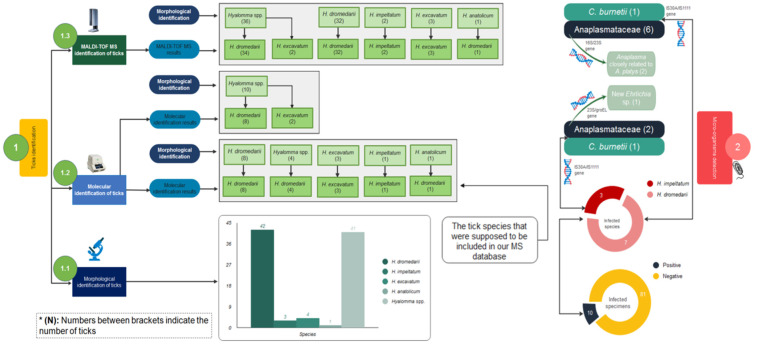
Study design showing two sections: tick identification and microorganism identification. The first section is summarized by a flowchart including morphological identification results (1.1), molecular identification results (1.2) and MALDI-TOF MS analysis results (1.3). The second section includes qPCR and sequencing results using different genes.

**Figure 5 microorganisms-10-02178-f005:**
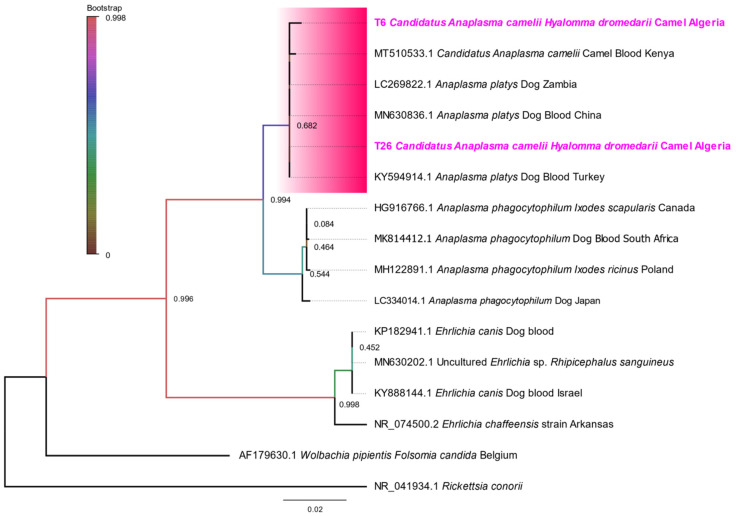
Representative maximum likelihood tree based on *16S* rRNA sequences using Tamura 3-parameter model. Statistical support for internal branches of the trees was evaluated by bootstrapping with 500 iterations. The sequences obtained are highlighted in bold, and the cluster of Ca. A Camelii and *A. platys* is colored in rose. *Rickettsia conorii* was used as an outgroup.

**Figure 6 microorganisms-10-02178-f006:**
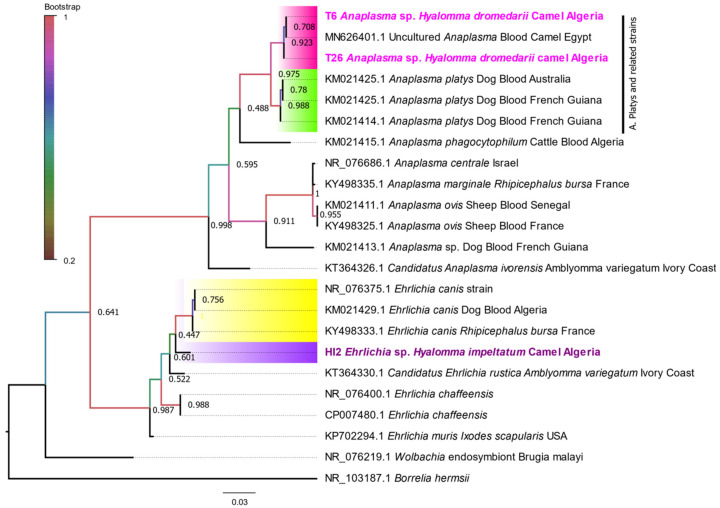
Phylogenetic tree of the *23S rRNA* gene using the maximum likelihood method. The values on the branches are bootstrap support values based on 500 replications. The position of the obtained sequence is indicated in bold. The subcluster of *A. platys* is colored in green, and the subcluster of the sequences obtained is colored in rose. The subcluster of the obtained sequence of *Ehrlichia* is colored in purple. *Borrelia hermsii* was used as an outgroup.

**Figure 7 microorganisms-10-02178-f007:**
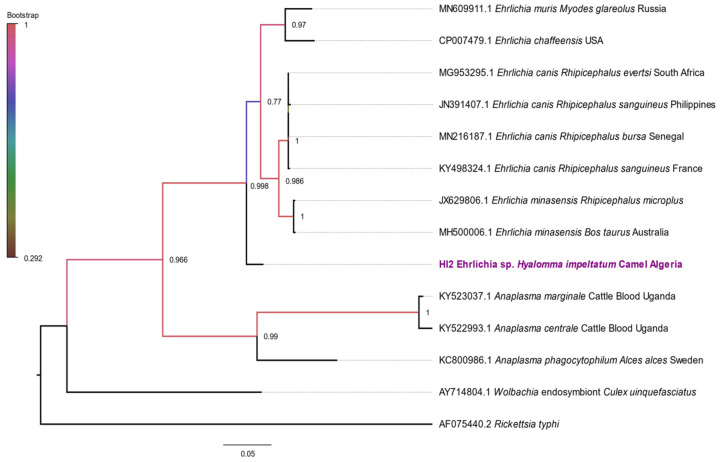
Phylogenetic tree based on *groEL* sequences aligned with MEGA 7 using Tamura 3-parameter model. Statistical support for internal branches of the trees was evaluated by bootstrapping with 500 iterations. *Rickettsia typhi* was added as an outgroup.

**Table 1 microorganisms-10-02178-t001:** Ticks used for the creation of MALDI-TOF MS reference database and blind tests.

Morphological ID	No. of Ticks	No. of Males	No. of Females	Good-Quality MS Spectra	Molecular ID *n* = Sequence GenBank ID %	No MS Reference Spectra Created	Blind Test	MS Blind Test ID	Log-Score Value (LSV) [Low-High]
*Hyalomma* spp. ***	41	0	41	40	*H. dromedarii* (*n* = 12) *H. excavatum* (*n* = 2) 99.52–100%	4	36/36	*H. dromedarii* (34)	1.890–2.507
								*H. excavatum* (2)	1.801–1.880
*H. dromedarii*	42	33	9	40	*H. dromedarii* (*n* = 8) 99.76–100%	8	32/32	*H. dromedarii*	1.701–2.431
*H. impeltatum*	3	3	-	3	*H. impeltatum* (*n* = 1) 99.75%	1	2/2	*H. impeltatum*	2.081–2.083
*H. excavatum*	4	4	-	4	*H. excavatum* (*n* = 3) 100%	1	3/3	*H. excavatum*	2.046–2.204
*H. anatolicum*	1	1	-	1	*H. dromedarii* (*n* = 1) 100%	-	1/1	*H. dromedarii*	2.342
Total	91	41	50	88	27	14	74/74	-	-

* Engorged specimens.

**Table 2 microorganisms-10-02178-t002:** Molecular characterization of Anaplasmataceae identified in ticks.

Tick Species	Host	*16SrRNA* Gene (*n* = Sequence; % ID)	*23S rRNA* Gene (*n* = Sequence; % ID)	*groEL* Gene (*n* = Sequence; % ID)
*Hyalomma dromedarii*	*Camelus dromedarius*	*Candidatus* A. camelii (*n* = 2; 100%)	*Anaplasma* sp. (*n* = 2; 99.78–100%)	-
*Hyalomma impeltatum*		*Candidatus* Midichloria mitochondrii (*n* = 1; 99.63%)	*Candidatus* Ehrlichia rustica (*n* = 1; 97.28%)	*E. Canis* (*n* = 1; 94%)

## Data Availability

The camel tick MALDI-TOF database is publicly accessible and can be downloaded with the following DOI number: https://doi.org/10.35081/srfb-n029, access on 1 December 2021. Tick *16S rRNA* sequences: *H. dromedarii*: OL672219, OL672220, OL672221, OL672222, OL672223 and OL672224; *H. impeltatum*: OL672225; *H. excavatum*: OL672226. Anaplasmataceae *16SrRNA* sequences: Uncultured *Anaplasma* sp.: OL672236 and OL672237; *Candidatus Midichloria mitochondrii*: OL672238. Anaplasmataceae *23S rRNA* sequences: Uncultured *Anaplasma* sp.: OL672216 and OL672217; uncultured *Ehrlichia* sp.: OL672218; *groEL* sequences: OL757650.

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
