# Peer review of "MALDI-TOF MS Identification of Dromedary Camel Ticks and Detection of Associated Microorganisms, Southern Algeria"

_microorganisms, 2022, doi:10.3390/microorganisms10112178_

Round 1
Reviewer 1 Report
The authors use mass spectrometry to identify species. Although this method has been used in the identification of other species, I think it is not a reasonable or advanced method. First of all, there are many proteins and small molecular substances in the species, which can reach tens of thousands. In terms of protein alone, it can reach tens of thousands. With more than 300 protein modifications, it is impossible to even estimate the type and number of proteins. In this way, tens of thousands of peaks will be displayed on the mass spectrum.
In addition, the best method for protein detection is to use the bottom up method. After enzymolysis, the fragment peptide is identified by the MS2. Therefore, MALDI-TOF is not suitable for protein identification if there is no HCD or CID for fragmentation, which has been a technology for more than 10 years ago.
Even if the samples collected by the author have no protein degradation, the ticks will be in different states, such as blood sucking period, such as high temperature, low temperature and other factors, and the protein in the body will change, so the results of mass spectrometry identification will be very different. Although it seems that the peaks of the same species in this paper are similar, this test will have great problems in the case of a large number of samples. It is far less accurate than gene cloning.
In this manuscriptr, mass spectrometry identification is considered as a novel point to conceive and write the paper, but I think this part should be removed, and species identification can be carried out completely through morphology and 16S. However, the content of this article is a little less and the novelty has declined after removing the mass spectrum identification part.
Author Response
Authors' response: Considering that the number of proteins in arthropod samples is extremely high, this can appear as a limitation of this technology and even raise doubts about its reliability for species identification. However, not all proteins are ionized when performing MALDI-TOF MS, and our spectrum is restricted to a certain biomolecule size. Therefore, although the protein content of the initial sample is very complex, we use a targeted and standardized method that allows us to meet our objective. This approach has been evaluated by our research team and others for the past ten years for the identification of arthropod vectors. We and others have shown and confirmed that MALDI-TOF MS is a reliable tool for arthropod identification. This work has been summarized in the review of Sevestre et al. 2021. This has now been discussed more extensively in the manuscript in lines 331-338 “As previously mentioned, the robustness of MALDI-TOF MS for arthropod identification has been evaluated for several years using laboratory reared and field collected samples. MALDI-TOF MS profiling uses a highly standardized protein extraction method coupled with spectrum acquisition approach to generate reference spectra that will be specific and reproducible enough to identify specimen of the same species [54]. MALDI-TOF MS has especially shown its reliability and usefulness in field studies by significantly decreasing the costs and time of identification but also allowing the detection of low occurring species [55].”
Reviewer 2 Report
The style of three tables is different, which should meet the requirement of journal.
Author Response
Authors' response: Thank you for your remark. The changes have been done as required.
Reviewer 3 Report
ligne 400:
-
Anaplasma sp.
-
Excellent work
Author Response
Authors' response: Thank you for your remark. We have added sp. after Anaplasma as follows in line 437 “Anaplasma sp.”
Round 2
Reviewer 1 Report
Although there have been published articles on the use of mass spectrometry for species identification before, they are not authoritative journals of taxonomy and cannot represent that the method is generally recognized by the scientific community. Whether proteins with similar molecular weights can be separated by using a high-resolution mass spectrometer, the whole protein mass spectrometry can not distinguish the arrangement differences of amino acids at all, that is, if two amino acids are exchanged, the enzymatic hydrolysis method and the secondary mass spectrometry fragmentation method are not used, the sequence of these two amino acids can not be distinguished at all. These will have a great impact on the results. Even if tof is cheaper than other mass spectrometers, general laboratories can't afford it. It is much more expensive than PCR. Taxonomy does not need rapid detection, but it needs accuracy. Compared with PCR, tof has obvious disadvantages, and the results are not credible. Therefore, the author insists on using this method for species identification, which will only bring confusion to taxonomy.